# Population Matching Discrepancy and Applications in Deep Learning

**Jianfei Chen,   Chongxuan Li,   Yizhong Ru,   Jun Zhu**[*]
Dept. of Comp. Sci. & Tech., TNList Lab, State Key Lab for Intell. Tech. & Sys.
Tsinghua University, Beijing, 100084, China
{chenjian14,licx14,ruyz13}@mails.tsinghua.edu.cn, dcszj@tsinghua.edu.cn

## Abstract

A differentiable estimation of the distance between two distributions based on samples is important for many deep learning tasks. One such estimation is maximum mean discrepancy (MMD). However, MMD suffers from its sensitive kernel bandwidth hyper-parameter, weak gradients, and large mini-batch size when used as a training objective. In this paper, we propose population matching discrepancy (PMD) for estimating the distribution distance based on samples, as well as an algorithm to learn the parameters of the distributions using PMD as an objective. PMD is defined as the minimum weight matching of sample populations from each distribution, and we prove that PMD is a strongly consistent estimator of the first Wasserstein metric. We apply PMD to two deep learning tasks, domain adaptation and generative modeling. Empirical results demonstrate that PMD overcomes the aforementioned drawbacks of MMD, and outperforms MMD on both tasks in terms of the performance as well as the convergence speed.

## 1   Introduction

Recent advances on image classification [26], speech recognition [19] and machine translation [9] suggest that properly building large models with a deep hierarchy can be effective to solve realistic learning problems. Many deep learning tasks, such as generative modeling [16, 3], domain adaptation [5, 47], model criticism [32] and metric learning [14], require estimating the statistical divergence of two probability distributions. A challenge is that in many tasks, only the samples instead of the closed-form distributions are available. Such distributions include implicit probability distributions and intractable marginal distributions. Without making explicit assumption on the parametric form, these distributions are richer and hence can lead to better estimates [35]. In these cases, the estimation of the statistical divergence based on samples is important. Furthermore, as the distance can be used as a training objective, it need to be differentiable with respect to the parameters of the distributions to enable efficient gradient-based training.

One popular sample-based statistical divergence is the maximum mean discrepancy (MMD) [17], which compares the kernel mean embedding of two distributions in RKHS. MMD has a closed-form estimate of the statistical distance in quadratic time, and there are theoretical results on bounding the approximation error. Due to its simplicity and theoretical guarantees, MMD have been widely adopted in many tasks such as belief propagation [44], domain adaptation [47] and generative modeling [31]. However, MMD has several drawbacks. For instance, it has a kernel bandwidth parameter that needs tuning [18], and the kernel can saturate so that the gradient vanishes [3] in a deep generative model. Furthermore, in order to have a reliable estimate of the distance, the mini-batch size must be large, e.g., 1000, which slows down the training by stochastic gradient descent [31].

---

[*]Corresponding author.

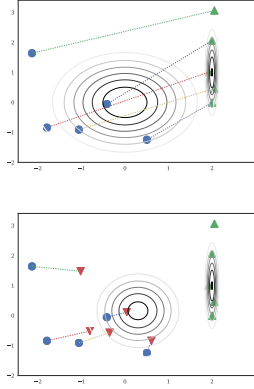

**Require:** Noise distributions $q_X, q_Y$ and transformations $T_{\theta_X}^X, T_{\theta_Y}^Y$.
Population size $N$, mini-batch size $|B|$.
**for** each iteration **do**
&emsp;Draw $\boldsymbol{\epsilon} \sim q_X(\cdot), \boldsymbol{\xi} \sim q_Y(\cdot)$
&emsp;Compute $x_{i;\theta_X} = T_{\theta_X}^X(\epsilon_i)$ and $y_{j;\theta_Y} = T_{\theta_Y}^Y(\xi_j)$
&emsp;$\mathbf{M} \leftarrow \text{MinimumWeightMatching}(\mathbf{X}_{\theta_X}, \mathbf{Y}_{\theta_Y})$
&emsp;Align the matched pairs $y_{1;\theta_Y}, \ldots, y_{N;\theta_Y} \leftarrow y_{M_1;\theta_Y}, \ldots, y_{M_N;\theta_Y}$
&emsp;**for** each mini batch $s \in [0, |B|, 2|B|, \ldots, N]$ **do**
&emsp;&emsp;$\boldsymbol{\theta} = \text{SGD}(\boldsymbol{\theta}, \frac{1}{|B|} \sum_{i=s}^{s+|B|-1} d(x_{i;\theta_X}, y_{i;\theta_Y}))$
&emsp;**end for**
**end for**

Figure 1: Pseudocode of PMD for parameter learning with graphical illustration of an iteration. Top: draw the populations and compute the matching; bottom: update the distribution parameters.

In this paper, we consider a sample-based estimation of the Wasserstein metric [49], which we refer to as population matching discrepancy (PMD). PMD is the cost of the minimum weight matching of the two sample populations from the distributions, and we show that it is a strongly consistent estimator of the first Wasserstein metric. We propose an algorithm to use PMD as a training objective to learn the parameters of the distribution, and reveal that PMD has some advantages over MMD: PMD has no bandwidth hyper-parameter, has stronger gradient, and can use normal mini-batch size, such as 100, during the learning. We compare PMD with MMD on two deep learning tasks, domain adaptation and generative modeling. PMD outperforms MMD in terms of both the performance and the speed of convergence.

## 2 Population Matching Discrepancy

In this section, we give the definition of the population matching discrepancy (PMD) and propose an algorithm to learn with PMD.

### 2.1 Population Matching Discrepancy

Consider the general case where we have two distributions $p_X(x)$ and $p_Y(y)$, whose PDFs are unknown, but we are allowed to draw samples from them. Let $\mathbf{X} = \{x_i\}_{i=1}^N$ and $\mathbf{Y} = \{y_j\}_{j=1}^N$ denote the $N$ i.i.d. samples from each distribution respectively. We define the $N$-PMD of the two distributions as

$$D_N(\mathbf{X}, \mathbf{Y}) = \min_{\mathbf{M}} \frac{1}{N} \sum_{i=1}^N d(x_i, y_{M_i}), \tag{1}$$

where $d(\cdot, \cdot)$ is any distance in the sample space (e.g., Euclidean distance) and $\mathbf{M}$ is a permutation to derive a matching between the two sets of samples. The optimal $\mathbf{M}$ corresponds to the bipartite minimum weight matching [27], where each element of the cost matrix is $d_{ij} = d(x_i, y_j)$ with $i, j \in [N]$, where $[N] = \{1, \cdots, N\}$. Intuitively, PMD is the average distance of the matched pairs of samples, therefore it is non-negative and symmetric. Furthermore, as we shall see in Sec. 3.1, PMD is a strongly consistent estimator of the first Wasserstein metric [49] between $p_X$ and $p_Y$, which is a valid statistical distance, i.e., $D_\infty(\mathbf{X}, \mathbf{Y}) = 0$ iff the two distributions $p_X$ and $p_Y$ are identical.

### 2.2 Parameter Learning

While the $N$-PMD in Eq. (1) itself can serve as a measure of the closeness of two distributions, we are more interested in learning the parameter of the distributions using PMD as an objective. For instance, in generative modeling [31], we have a parameterized generator distribution $p_X(x; \theta_X)$ and a data distribution $p_Y(y)$, and we wish to minimize the distance of these two distributions. We

assume the samples are obtained by applying some parameterized transformations to a known and fixed noise distribution, i.e.,

$$\epsilon_i \sim q_X(\epsilon), \ x_{i;\theta_X} = T^X_{\theta_X}(\epsilon_i); \text{ and } \xi_j \sim q_Y(\xi), \ y_{j;\theta_Y} = T^Y_{\theta_Y}(\xi_j).$$

For flexibility, the transformations can be implemented by deep neural networks. Without loss of generality, we assume both $p_X$ and $p_Y$ are parameterized distributions by $\theta_X$ and $\theta_Y$, respectively. If $p_X$ is a fixed distribution, we can take $q_X = p_X$ and $T^X_{\theta_X}$ to be a fixed identity mapping. Our goal for parameter learning is to minimize the expected $N$-PMD over different populations

$$\min_{\theta_X,\theta_Y} \mathbb{E}_{\epsilon,\xi} D_N(\mathbf{X}_{\theta_X}, \mathbf{Y}_{\theta_Y}), \tag{2}$$

where $\epsilon = \{\epsilon_i\}_{i=1}^N$, $\xi = \{\xi_j\}_{j=1}^N$, $\mathbf{X}_{\theta_X} = \{x_{i;\theta_X}\}_{i=1}^N$ and $\mathbf{Y}_{\theta_Y} = \{y_{j;\theta_Y}\}_{j=1}^N$, and the expectation is for preventing over-fitting the parameter with respect to particular populations. The parameters can be optimized by stochastic gradient descent (SGD) [7]. At each iteration, we draw $\epsilon$ and $\xi$, and compute an unbiased stochastic gradient

$$\nabla_{\boldsymbol{\theta}} D_N(\mathbf{X}_{\theta_X}, \mathbf{Y}_{\theta_Y}) = \nabla_{\boldsymbol{\theta}} \min_{\mathbf{M}} \frac{1}{N} \sum_{i=1}^N d(x_{i;\theta_X}, y_{M_i;\theta_Y}) = \nabla_{\boldsymbol{\theta}} \frac{1}{N} \sum_{i=1}^N d(x_{i;\theta_X}, y_{M_i^*;\theta_Y}), \tag{3}$$

where $\mathbf{M}^* = \operatorname{argmin}_{\mathbf{M}} \sum_{i=1}^N d(x_{i;\theta_X}, y_{M_i;\theta_Y})$ is the minimum weight matching for $\mathbf{X}_{\theta_X}$ and $\mathbf{Y}_{\theta_Y}$. The second equality in Eq. (3) holds because the *discrete* matching $\mathbf{M}^*$ should not change for infinitesimal change of $\boldsymbol{\theta}$, as long as the transformations $T^X$, $T^Y$, and the distance $d(\cdot,\cdot)$ are continuous. In other words, the gradient does not propagate through the matching.

Furthermore, assuming that the matching $\mathbf{M}^*$ does not change much within a small number of gradient updates, we can have an even cheaper stochastic gradient by subsampling the populations

$$\nabla_{\boldsymbol{\theta}} D_N(\mathbf{X}_{\theta_X}, \mathbf{Y}_{\theta_Y}) \approx \nabla_{\boldsymbol{\theta}} \frac{1}{|B|} \sum_{i=1}^{|B|} d(x_{B_i;\theta_X}, y_{M_{B_i}^*;\theta_Y}), \tag{4}$$

where a mini-batch of $|B|$, e.g., 100, samples is used to approximate the whole $N$-sample population. To clarify, our population size $N$ is known as the mini-batch size in some maximum mean discrepancy (MMD) literature [31], and is around 1000. Fig. 1 is the pseudocode of parameter learning for PMD along with a graphical illustration. In the outer loop, we generate populations and compute the matching; and in the inner loop, we perform several SGD updates of the parameter $\boldsymbol{\theta}$, assuming the matching $\mathbf{M}$ does not change much. In the graphical illustration, the distribution $p_Y$ is fixed, and we want to optimize the parameters of $p_X$ to minimize their PMD.

## 2.3 Solving the Matching Problem

The minimum weight matching can be solved exactly in $O(N^3)$ by the Hungarian algorithm [27]. When the problem is simple enough, so that small $N$, e.g., hundreds, is sufficient for reliable distance estimation, $O(N^3)$ time complexity is acceptable comparing with the $O(N \times \text{BackProp})$ time complexity of computing the gradient with respect to the transformations $T^X_{\theta_X}$ and $T^Y_{\theta_Y}$. When $N$ is larger, e.g., a few thousands, the Hungarian algorithm takes seconds to run. We resort to Drake and Hougardy's approximated matching algorithm [11] in $O(N^2)$ time. The running time and model quality of PMD using both matching algorithms are reported in Sec. 5.3. In practice, we find PMD with both the exact and approximate matching algorithms works well. This is not surprising because training each sample towards its approximate matching sample is still reasonable. Finally, while we only implement the serial CPU version of the matching algorithms, both algorithm can be parallelized on GPU to further improve the running speed [10, 34].

## 3 Theoretical Analysis and Connections to Other Discrepancies

In this section, we establish the connection between PMD with the Wasserstein metric and the maximum mean discrepancy (MMD). We show that PMD is a strongly consistent estimator of the Wasserstein metric, and compare its advantages and disadvantages with MMD.

## 3.1 Relationship with the Wasserstein Metric

The Wasserstein metric [49] was initially studied in the optimal transport theory, and has been adopted in computer vision [40], information retrieval [50] and differential privacy [30]. The first Wasserstein metric of two distributions $p_X(x)$ and $p_Y(y)$ is defined as

$$
\inf_{\gamma(x,y)} \int d(x,y)\gamma(x,y)\mathrm{d}x\mathrm{d}y
$$

$$
\text{s.t.} \int \gamma(x,y)\mathrm{d}x = p_Y(y), \forall y; \int \gamma(x,y)\mathrm{d}y = p_X(x), \forall x; \gamma(x,y) \geq 0, \forall x,y. \tag{5}
$$

Intuitively, the Wasserstein metric is the optimal cost to move some mass distributed as $p_X$ to $p_Y$, where the transference plan $\gamma(x,y)$ is the amount of mass to move from $x$ to $y$. Problem (5) is not tractable because the PDFs of $p_X$ and $p_Y$ are unknown. We approximate them with empirical distributions $\hat{p}_X(x) = \frac{1}{N}\sum_{i=1}^{N}\delta_{x_i}(x)$ and $\hat{p}_Y(y) = \frac{1}{N}\sum_{j=1}^{N}\delta_{y_j}(y)$, where $\delta_x(\cdot)$ is the Dirac delta function at $x$. To satisfy the constraints, $\gamma$ should have the form $\gamma(x,y) = \sum_{i=1}^{N}\sum_{j=1}^{N}\gamma_{ij}\delta_{x_i,y_j}(x,y)$, where $\gamma_{ij} \geq 0$. Letting $p_X = \hat{p}_X$ and $p_Y = \hat{p}_Y$, we can simplify problem (5) as follows

$$
\min_{\gamma} \sum_{i=1}^{N}\sum_{j=1}^{N} d(x_i,y_j)\gamma_{ij} \quad \text{s.t.} \sum_{j=1}^{N}\gamma_{ij} = \frac{1}{N}, i \in [N]; \sum_{i=1}^{N}\gamma_{ij} = \frac{1}{N}, j \in [N]; \gamma_{ij} \geq 0. \tag{6}
$$

The linear program (6) is equivalent to the minimum weight matching problem [27], i.e., there exists a permutation $M_1, \ldots, M_N$, such that $\gamma(x_i, y_{M_i}) = \frac{1}{N}$ is an optimal solution (see Proposition 5.4 in [6]). Plugging such $\gamma$ back to problem (6), we obtain Eq. (1), the original definition of PMD.

Furthermore, we can show that the solution of problem (6), i.e., the $N$-PMD, is a strongly consistent estimator of the first Wasserstein metric in problem (5).

**Definition 1** (Weak Convergence of Measure [48]). *A sequence of probability distributions $p_N, N = 1, 2, \ldots$ converges weakly to the probability distribution $p$, denoted as $p_n \Rightarrow p$, if $\lim_{N\to\infty} \mathbb{E}_{p_N}[f] = \mathbb{E}_p[f]$ for all bounded continuous functions $f$.*

**Proposition 3.1** (Varadarajan Theorem [48]). *Let $x_1, \ldots, x_N, \ldots$ be independent, identically distributed real random variables with the density function $p(x)$, let $p_N(x) = \frac{1}{N}\sum_{i=1}^{N}\delta_{x_N}(x)$ where $\delta_{x_N}(\cdot)$ is the Dirac delta function. Then $p_N \Rightarrow p$ almost surely.*

**Proposition 3.2** (Stability of Optimal Transport [49]). *Let $\mathcal{X}$ and $\mathcal{Y}$ be Polish spaces and let $d : \mathcal{X} \times \mathcal{Y} \to \mathbb{R}$ be a continuous function s.t. $\inf d > -\infty$. Let $\{p_N^X\}_{N\in\mathbb{N}}$ and $\{p_N^Y\}_{N\in\mathbb{N}}$ be sequences of probability distributions on $\mathcal{X}$ and $\mathcal{Y}$ respectively. Assume that $p_N^X \Rightarrow p_X$ (resp. $p_N^Y \Rightarrow p_Y$). For each $N$, let $\gamma_N$ be an optimal transference plan between $p_N^X$ and $p_N^Y$. If $\liminf_{N\in\mathbb{N}} \int d(x,y)\gamma_N(x,y)dxdy < +\infty$, then $\gamma_N \Rightarrow \gamma$, where $\gamma$ is an optimal transference plan between $p_X$ and $p_Y$.*

Proposition 3.2 is a special case of Theorem 5.20 in [49] with fixed function $d$. The following theorem is the main result of this section.

**Theorem 3.3** (Strong Consistency of PMD). *Let $x_1, \ldots, x_N, \ldots$ and $y_1, \ldots, y_N, \ldots$ be independent, identically distributed real random variables from $p_X$ and $p_Y$, respectively. We construct a sequence of PMD problems (6) between $p_N^X(x) = \frac{1}{N}\sum_{i=1}^{N}\delta_{x_N}(x)$ and $p_N^Y(y) = \frac{1}{N}\sum_{i=1}^{N}\delta_{y_N}(y)$. Let $\gamma_N$ be the optimal transference plan of the $N$-th PMD problem. Then the sequence $\gamma_N \Rightarrow \gamma$ almost surely, where $\gamma$ is the optimal transference plan between $p_X$ and $p_Y$. Moreover, $\lim_{N\to\infty} \int d(x,y)\gamma_N(x,y)dxdy = \int d(x,y)\gamma(x,y)dxdy$ almost surely.*

The proof is straightforward by applying Proposition 3.1 and 3.2. We also perform an empirical study of the approximation error with respect to the population size in Fig. 2(a).

While the Wasserstein metric has been widely adopted in various machine learning and data mining tasks [40, 50, 30], it is usually used to measure the similarity between two discrete distributions, e.g., histograms. In contrast, PMD is a stochastic approximation of the Wasserstein metric between two continuous distributions. There is also work on estimating the Wasserstein metric of continuous distributions based on samples [45]. Unlike PMD, which is approximating the primal problem, they approximate the dual. Their approximation is not differentiable with respect to the distribution

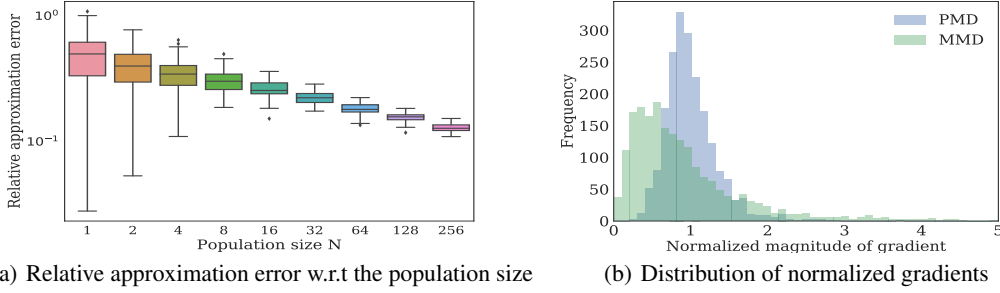

(a) Relative approximation error w.r.t the population size  (b) Distribution of normalized gradients

Figure 2: Some empirical analysis results. The detailed experiment setting is described in Sec. 5.4.

parameters because the parameters appear in the constraint instead of the objective. Recently, Wasserstein GAN (WGAN) [3] proposes approximating the dual Wasserstein metric by using a neural network "critic" in place of a 1-Lipschitz function. While WGAN has shown excellent performance on generative modeling, it can only compute a relative value of the Wasserstein metric upon to an unknown scale factor depending on the Lipschitz constant of the critic neural network. PMD also differs from WGAN by not requiring a separate critic network with additional parameters. Instead, PMD is parameter free and can be computed in polynomial time.

## 3.2 Relationship with MMD

Maximum mean discrepancy (MMD) [17] is a popular method for estimating the distance between two distributions by samples, defined as follows

$$D_{MMD}(\mathbf{X}, \mathbf{Y}) = \frac{1}{N^2} \sum_{i=1}^{N} \sum_{j=1}^{N} k(x_i, x_j) - \frac{2}{NM} \sum_{i=1}^{N} \sum_{j=1}^{M} k(x_i, y_j) + \frac{1}{M^2} \sum_{i=1}^{M} \sum_{j=1}^{M} k(y_i, y_j),$$

where $k(\cdot, \cdot)$ is a kernel, e.g., $k(x, y) = \exp(-\|x - y\|^2 / 2\sigma^2)$ is the RBF kernel with bandwidth $\sigma$. Both MMD and the Wasserstein metric are integral probability metrics [17], with different function classes. MMD has a closed-form objective, and can be evaluated in $O(NMD)$ if $x$ and $y$ are $D$-dimensional vectors. In contrast, PMD needs to solve a matching problem, and the time complexity is $O(N^2 D)$ for computing the distance matrix, $O(N^3)$ for exact Hungarian matching, and $O(N^2)$ for approximated matching. However, as we argued in Sec. 2.3, the time complexity for computing matching is still acceptable comparing with the cost of training neural networks.

Comparing with MMD, PMD has a number of advantages:

**Fewer hyper-parameter** PMD do not have the kernel bandwidth $\sigma$, which needs tuning.

**Stronger gradient** Using the RBF kernel, the gradient of MMD w.r.t a particular sample $x_i$ is $\nabla_{x_i} D_{MMD}(\mathbf{X}, \mathbf{Y}) = \frac{1}{N^2} \sum_j k(x_i, x_j) \frac{x_j - x_i}{\sigma^2} - \frac{2}{NM} \sum_j k(x_i, y_j) \frac{y_j - x_i}{\sigma^2}$. When minimizing MMD, the first term is a repulsive term between the samples from $p_X$, and the second term is an attractive term between the samples from $p_X$ and $p_Y$. The L2 norm of the term between two samples $x$ and $y$ is $k(x, y) \frac{\|x - y\|_2}{\sigma^2}$, which is small if $\|x - y\|_2$ is either too small or too large. As a result, if a sample $x_i$ is an outlier, i.e., it is not close to any samples from $p_Y$, all the $k(x_i, y_j)$ terms are small and $x_i$ will not receive strong gradients. On the other hand, if all the samples $x_i, i \in [N]$ are close to each other, $x_j - x_i$ is small, so that repulsive term of the gradient is weak. Both cases slow down the training. In contrast, if $d(x, y) = |x - y|$ is the L1 distance, the gradient of PMD $\nabla_{x_i} D_N(\mathbf{X}, \mathbf{Y}) = \frac{1}{N} \text{sgn}(x_i - y_{M_i})$, where $\text{sgn}(\cdot)$ is the sign function, is always strong regardless of the closeness between $x_i$ and $y_{M_i}$. We compare the distribution of the relative magnitude of the gradient of the parameters contributed by each sample in Fig. 2(b). The PMD gradients have similar magnitude for each sample, while there are many samples have small gradients for MMD.

**Smaller mini-batch size** As we see in Sec 2.2, the SGD mini-batch size for PMD can be smaller than the population size; while the mini-batch size for MMD must be equal with the population size. This is because PMD only considers the distance between a sample and its matched sample, while

MMD considers the distance between all pairs of samples. As the result of smaller mini-batch size, PMD can converge faster than MMD when used as a training objective.

# 4 Applications

## 4.1 Domain Adaptation

Now we consider a scenario where the labeled data is scarce in some domain of interest (target domain) but that is abundant in some related domain (source domain). Assuming that the data distribution $p_S(X, y)$ for the source domain and that of the target domain, i.e. $p_T(X, y)$ are similar but not the same, unsupervised domain adaptation aims to train a model for the target domain, given some labeled data $\{(X_i^S, y_i^S)\}_{i=1}^{N_S}$ from the source domain and some unlabeled data $\{X_j^T\}_{j=1}^{N_T}$ from the target domain. According to the domain adaptation theory [5], the generalization error on the target domain depends on the generalization error on the source domain as well as the difference between the two domains. Therefore, one possible solution for domain adaptation is to learn a feature extractor $\phi(X)$ shared by both domains, which defines feature distributions $p_S^\phi$ and $p_T^\phi$ for both domains, and minimize some distance between the feature distributions [47] as a regularization. Since the data distribution is inaccessible, we replace all distributions with their empirical distributions $\hat{p}_S$, $\hat{p}_T$, $\hat{p}_S^\phi$ and $\hat{p}_T^\phi$, and the training objective is

$$\mathbb{E}_{X,y\sim\hat{p}_S}\mathcal{L}(y, h(\phi(X))) + \lambda D(\hat{p}_S^\phi, \hat{p}_T^\phi),$$

where $\mathcal{L}(\cdot, \cdot)$ is a loss function, $h(\cdot)$ is a classifier, $\lambda$ is a hyper-parameter, and $D(\hat{p}_S^\phi, \hat{p}_T^\phi)$ is the domain adaptation regularization. While the Wasserstein metric itself of two empirical distribution is tractable, it can be too expensive to compute due to the large size of the dataset. Therefore, we still approximate the distance with (expected) PMD, i.e., $D(\hat{p}_S^\phi, \hat{p}_T^\phi) \approx \mathbb{E}_{\mathbf{X}^S\sim\hat{p}_S, \mathbf{X}^T\sim\hat{p}_T} D_{PMD}(\phi(\mathbf{X}^S), \phi(\mathbf{X}^T))$.

## 4.2 Deep Generative Modeling

Deep generative models (DGMs) aim at capturing the complex structures of the data by combining hierarchical architectures and probabilistic modelling. They have been proven effective on image generation [38] and semi-supervised learning [23] recently. There are many different DGMs, including tractable auto-regressive models [37], latent variable models [24, 39], and implicit probabilistic models [16, 31]. We focus on learning implicit probabilistic models, which define probability distributions on sample space flexibly without a closed-form. However, as described in Sec. 2.2, we can draw samples $X = T_{\theta_X}^X(\epsilon)$ efficiently from the models by transforming a random noise $\epsilon \sim q(\epsilon)$, where $q$ is a simple distribution (e.g. uniform), to $X$ through a parameterized model (e.g. neural network). The parameters in the models are trained to minimize some distance between the model distribution $p_X(X)$ and the empirical data distribution $\hat{p}_Y(Y)$. The distance can be defined based on an parameterized adversary, i.e., another neural network [16, 3], or directly with the samples [31]. We choose the distance to be the first Wasserstein metric, and employ its finite-sample estimator (i.e., the $N$-PMD defined in Eq. (2)) as training objective directly. Training this model with MMD is known as generative moment matching networks [31, 12].

# 5 Experiments

We now study the empirical performance of PMD and compare it with MMD. In the experiments, PMD always use the L1 distance, and MMD always use the RBF kernel. Our experiment is conducted on a machine with Nvidia Titan X (Pascal) GPU and Intel E5-2683v3 CPU. We implement the models in TensorFlow [1]. The matching algorithms are implemented in C++ with a single thread, and we write a CUDA kernel for computing the all-pair L1 distance within a population. The CUDA program is compiled with `nvcc` 8.0 and the C++ program is compiled with g++ 4.8.4, while -O3 flag is used for both programs. We use the approximate matching for the generative modeling experiment and exact Hungarian matching for all the other experiments.

## 5.1 Domain Adaptation

We compare the performance of PMD and MMD on the standard Office [41] object recognition benchmark for domain adaptation. The dataset contains three domains: amazon, dslr and webcam, and

Table 1: All the 6 unsupervised domain adaptation accuracy on the Office dataset between the amazon (a), dslr (d) and webcam (w) domains, in percentage. SVM and NN are trained only on the source domain, where NN uses the same architecture of PMD and MMD, but set $\lambda = 0$.

| Method | $a \to w$ | $d \to w$ | $w \to d$ | $a \to d$ | $d \to a$ | $w \to a$ | avg. |
|---|---|---|---|---|---|---|---|
| DDC [47] | 59.4±.8 | 92.5±.3 | 91.7±.8 | - | - | - | - |
| DANN [13] | 73.0 | **96**.4 | 99.2 | - | - | - | - |
| CMD [52] | 77.0±.6 | 96.3±.4 | 99.2±.2 | 79.6 ± .6 | 63.8±.7 | 63.3±.6 | 79.9 |
| JAN-xy [33] | 78.1±.4 | **96**.4 ± .2 | 99.3±.1 | 77.5 ± .2 | **68**.4 ± .2 | 65.0±.4 | 80.8 |
| SVM | 65.0 | 96.1 | 99.4 | 70.7 | 56.4 | 55.1 | 73.8 |
| NN | 67.8±.5 | 96.3±.2 | 99.5±.2 | 73.9 ± .6 | 58.5±.3 | 58.1±.3 | 75.7 |
| MMD | 76.9±.8 | 96.2±.2 | **99.6 ± .2** | 78.4±1.0 | 64.9±.5 | **68.1 ± .6** | 80.7 |
| PMD | **86.2 ± .7** | 96.2±.3 | 99.5±.2 | **82.7 ± .8** | 64.3±.4 | 66.8±.4 | **82.6** |

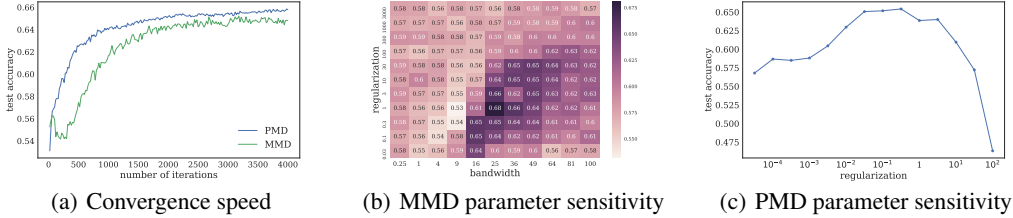

(a) Convergence speed    (b) MMD parameter sensitivity    (c) PMD parameter sensitivity

Figure 3: Convergence speed and parameter sensitivity on the Office $d \to a$ task.

there are 31 classes. Following [52], we use the 4096-dimensional VGG-16 [43] feature pretrained on ImageNet as the input. The classifier is a fully-connected neural network with a single hidden layer of 256 ReLU [15] units, trained with AdaDelta [51]. The domain regularization term is put on the hidden layer. We apply batch normalization [21] on the hidden layer, and the activations from the source and the target domain are normalized separately. Following [8], we validate the domain regularization strength $\lambda$ and the MMD kernel bandwidth $\sigma$ on a random 100-sample labeled dataset on the target domain, but the model is trained without any labeled data from the target domain. The experiment is then repeated for 10 times on the hyper-parameters with the best validation error. Since we perform such validation for both PMD and MMD, the comparison between them is fair. The result is reported in Table 1, and PMD outperforms MMD on the $a \to w$ and $a \to d$ tasks by a large margin, and is comparable with MMD on the other 4 tasks.

Then, we compare the convergence speed of PMD and MMD on the $d \to a$ task. We choose this task because PMD and MMD have similar performance on it. The result is shown in Fig. 3(a), where PMD converges faster than MMD. We also show the parameter sensitivity of MMD and PMD as Fig. 3(b) and Fig. 3(c), respectively. The performance of MMD is sensitive to both the regularization parameter $\lambda$ and the kernel bandwidth $\sigma$, so we need to tune both parameters. In contrast, PMD only has one parameter to tune.

## 5.2 Generative Modeling

We compare PMD with MMD for image generation on the MNIST [28], SVHN [36] and LFW [20] dataset. For SVHN, we train the models on the 73257-image training set. The LFW dataset is converted to $32 \times 32$ gray-scale images [2], and there are 13233 images for training. The noise $\epsilon$ follows a uniform distribution $[-1, 1]^{40}$. We implemented three architectures, including a fully-connected (fc) network as the transformation $T_{\theta_X}^X$, a deconvolutional (conv) network, and a fully-connected network for generating the auto-encoder codes (ae) [31], where the auto-encoder is a convolutional one pre-trained on the dataset. For MMD, we use a mixture of kernels of different bandwidths for the fc and conv architecture, and the bandwidth is fixed at 1 for the ae architecture, following the settings in the generative moment matching networks (GMMN) paper. We set the population size $N = 2000$ for both PMD and MMD, and the mini-batch size $|B| = 100$ for PMD. We use the AdaM optimizer [22] with batch normalization [21], and train the model for 100 epochs for PMD, and 500 epoches for MMD. The generated images on the SVHN and LFW dataset are

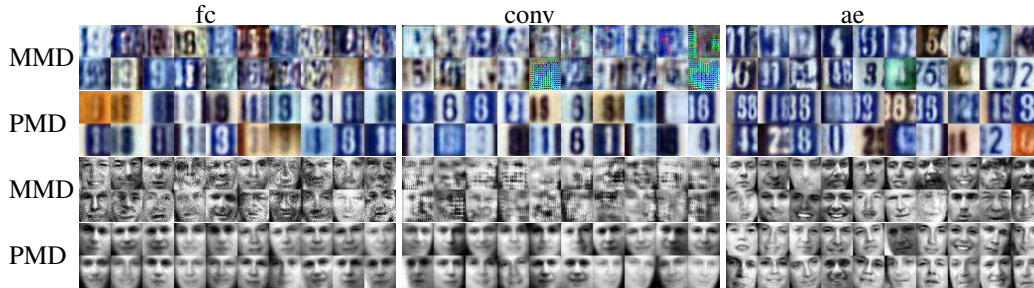

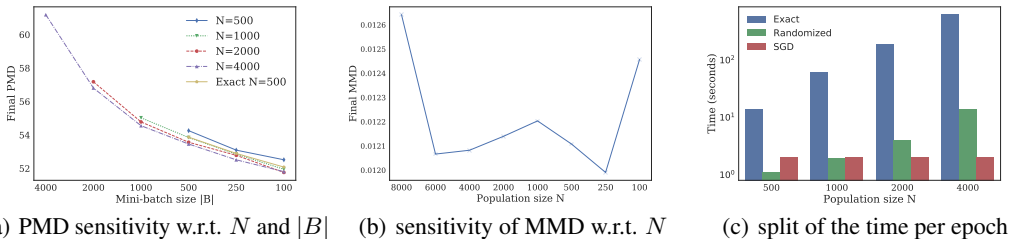

Figure 4: Image generation results on SVHN (top two rows) and LFW (bottom two rows).

(a) PMD sensitivity w.r.t. $N$ and $|B|$    (b) sensitivity of MMD w.r.t. $N$    (c) split of the time per epoch

Figure 5: Convergence and timing results. The "Exact $N = 500$" curve in (a) uses the Hungarian algorithm, and the rest uses the approximated matching algorithm.

presented in Fig. 4, and the images on the MNIST dataset can be found in the supplementary material. We observe that the images generated by PMD are less noisy than that generated by MMD. While MMD only performs well on the autoencoder code space (ae), PMD generates acceptable images on pixel space. We also noticed the generated images of PMD on the SVHN and LFW datasets are blurry. One reason for this is the pixel-level L1 distance is not good for natural images. Therefore, learning the generative model on the code space helps. To verify that PMD does not trivially reproduce the training dataset, we perform a circular interpolation in the representation space $q(\epsilon)$ between 5 random points, the result is available in the supplementary material.

## 5.3 Convergence Speed and Time Consumption

We study the impact of the population size $N$, the mini-batch size $|B|$ and the choice of matching algorithm to PMD. Fig. 5(a) shows the final PMD evaluated on $N = 2000$ samples on the MNIST dataset, using the fc architecture, after 100 epoches. The results show that the solution is insensitive to neither the population size $N$ nor the choice of the matching algorithm, which implies that we can use the cheap approximated matching and relatively small population size for speed. On the other hand, decreasing the mini-batch size $|B|$ improves the final PMD significantly, supporting our claim in Sec. 3.2 that the ability of using small $|B|$ is indeed an advantage for PMD. Unlike PMD, there is a trade-off for selecting the population size $N$ for MMD, as shown in Fig. 5(b). If $N$ is too large, the SGD optimization converges slowly; if $N$ is too small, the MMD estimation is unreliable. Fig. 5(c) shows the total time spent on exact matching, approximated matching and SGD respectively for each epoch. The cost of approximated matching is comparable with the cost of SGD. Again, we emphasize while we only have single thread implementations for the matching algorithms, both the exact [10] and approximated matching [34] can be significantly accelerated with GPU.

## 5.4 Empirical Studies

We examine the approximation error of PMD on a toy dataset. We compute the distances between two 5-dimensional standard isotropic Gaussian distributions. One distribution is centered at the origin and the other is at $(10, 0, 0, 0, 0)$. The first Wasserstein metric between these two distributions is 10. We vary the population size $N$ and compute the relative approximation error $= |D_N(\mathbf{X}, \mathbf{Y}) - 10|/10$ for 100 different populations $(\mathbf{X}, \mathbf{Y})$ for each $N$. The result is shown in Fig. 2(a). We perform a

linear regression between $\log N$ and the logarithm of expected approximation error, and find that the error is roughly proportional to $N^{-0.23}$.

We also validate the claim in Sec. 3.2 on the stronger gradients of PMD than that of MMD. We calculate the magnitude (in L2 norm) of the gradient of the parameters contributed by each sample. The gradients are computed on the converged model, and the model is the same as Sec. 5.3. Because the scale of the gradients depend on the scale of the loss function, we normalize the magnitudes by dividing them with the average magnitude of the gradients. We then show the distribution of normalized magnitudes of gradients in Fig. 2(b). The PMD gradients contributed by each sample are close with each other, while there are many samples contributing small gradients for MMD, which may slow down the fitting of these samples.

## 6    Conclusions

We present population matching discrepancy (PMD) for estimating the distance between two probability distributions by samples. PMD is the minimum weight matching between two random populations from the distributions, and we show that PMD is a strongly consistent estimator of the first Wasserstein metric. We also propose a stochastic gradient descent algorithm to learn parameters of the distributions using PMD. Comparing with the popular maximum mean discrepancy (MMD), PMD has no kernel bandwidth hyper-parameter, stronger gradient and smaller mini-batch size for gradient-based optimization. We apply PMD to domain adaptation and generative modeling tasks. Empirical results show that PMD outperforms MMD in terms of performance and convergence speed in both tasks. In the future, we plan to derive finite-sample error bounds for PMD, study its testing power, and accelerate the computation of minimum weight matching with GPU.

### Acknowledgments

This work is supported by the National NSF of China (Nos. 61620106010, 61621136008, 61332007), the MIIT Grant of Int. Man. Comp. Stan (No. 2016ZXFB00001), the Youth Top-notch Talent Support Program, Tsinghua Tiangong Institute for Intelligent Computing and the NVIDIA NVAIL Program.

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
