[Supplementary Material · Population_Matching_Discrepancy__arXiv_(4)_supp.pdf]

# A  Image generation results on MNIST

Figure 6: MNIST image generation. Top: fc, middle: conv, bottom: ae. Left: MMD, right: PMD.

Figure 7: Interpolation on the representation space using the convolutional generator. The first column is the randomly selected points and the rest are the interpolation between them.

# B  Adversarial learning of distances

From Fig. 4 we can see that the images generated by PMD is more blurry than those generated by generative adversarial networks (GANs) [16]. This is because the pixel-wise distance $d(x, y) = \|x - y\|_1$ is not a good distance between natural images. If we translate an image by one pixel, the pixel-wise distance between the two images will be much greater than zero. But the two images have identical semantic meanings, so the ideal distance between them should be close to zero.

Two recent works MMD GAN [29] and Cramer GAN [4] proposed to solve this problem by adversarially learn the distance between images. Inspired by these works, we present PMD GAN, which applies PMD for generative modeling with adversarially learned image distance. Following the discussion in Sec. 4.2, we formulate the problem of deep generative modeling as minimizing the PMD between a model distribution $p_{X;\theta}(x)$ and a data distribution $p_Y(y)$. We choose the image distance $d(x, y) = \sqrt{\|F(x) - F(y)\|_1}$ on the feature space defined by a feature extractor $F$. Following MMD GAN, we also define a decoder $D$ such that the reconstruction error between $x$ and $D(F(x))$ is minimized, to approximately enforce $F$ as a bijection. The overall objective is defined as

$$\min_{\theta} \max_{F} \left\{ \min_{\mathbf{M}} \frac{1}{N} \sum_{i=1}^{N} d(F(x_i), F(y_{M_i})) - \frac{\lambda}{N} \sum_{i=1}^{N} \left[ (x_i - D(F(x_i)))^2 + (y_i - D(F(y_i)))^2 \right] \right\},$$

where the feature extractor $F$ wants to maximize the PMD while minimizing the reconstruction error, and the generator $p_{X;\theta}$ wants to minimize the PMD.

We test the proposed approach on the CIFAR10 dataset [25]. We implement our model in Tensor-Flow [1] and make our implementation similar as the PyTorch implementation of MMD GAN. We compare our model with MMD GAN, which uses a mixture of RBF kernels with the bandwidth $[1, 2, 4, 8, 16]$. Both models uses a batch size of $64$ and RMSProp [46] optimizer with $5 \times 10^{-5}$ learning rate. We clip the weights in the range $[-0.01, 0.01]$ following Wasserstein GAN [3]. The auto encoder regularization parameter $\lambda$ is set to 8. We run PMD GAN, MMD GAN and Wasserstein GAN for 100 epochs, and the result are shown in Fig. 8. The results confirm that using adversarially learned distance, PMD can generate sharp images.

(a) MMD

(b) PMD

(c) WGAN

Figure 8: Image generation with adversarially learned distance. Inception score [42] are 4.50 for all the three models.

|  | $\mu_1$ | $\mu_2$ | $\log \sigma_1$ | $\log \sigma_2$ |
|---|---|---|---|---|
| Data | (8.34, 14.4) | (0, 6.05) | diag(0, 0) | diag(0, 0) |
| Learned | (8.17, 14.3) | (0.28, 6.30) | diag(-5.38, -5.53) | diag(-3.25, -4.16) |

Table 2: Mean and logarithm of standard deviation of the Gaussian mixture experiment.

## C  Handling multimodal distributions

It is an interesting question that whether the particle based methods (PMD or MMD) can handle distributions with more modes than the number of particles used. The answer is yes. We discuss a particular Gaussian mixture generation example. In this task, we have an unknown data distribution, which is a Gaussian mixture distribution with two mixing components $\mathcal{N}(\mu_1, \sigma_1^2)$ and $\mathcal{N}(\mu_2, \sigma_2^2)$. We want to learn its parameters $\mu_1$, $\mu_2$, $\sigma_1$ and $\sigma_2$ by minimizing the distance between the model distribution and the data distribution, using only *one* particle per gradient step. MMD can still learn the distribution because it is unbiased [17]. Empirically, PMD can also estimate the mean and variance, using a distance $d(x, y) = -\exp(-\|x - y\|_1 /\sigma)$, where $\sigma$ is a bandwidth. Table 2 shows the true model parameters and the parameters learned by PMD. Even with $N = 1$, PMD still estimates the mean fairly well, despite underestimating the variance.