[Reviews · NeurIPS 2017]

Reviewer 1



The paper defines Population Matching Discrepancy between two distributions as the Wasserstein distance between two minibatches from the distributions. The Wasserstein distance is computed by an exact O(N^3) or an approximate O(N^2) algorithm. Pros: - It is interesting to see the experiments with this computation of the Wasserstein distance. The generated images are not as good as from Wasserstein GAN. Cons: - The proposed distance would need large N to estimate the Wasserstein distance between two diverse multimodal distributions. I suspect that problems would be already visible, if trying to match a mixture of Gaussians (including learning the variances). - If N is not large enough, the optimization may have the global minimum at a point different from the true distribution. For example, the learned distribution may have less entropy. The SVHN samples in Figure 4 seem to have low diversity. Digit 8 appears frequently there. Minor typos: - Line 141: s/usally/usually/ Update: I have read the rebuttal. Thanks for the extra experiments. The authors should clarity the limitation of PMD and MMD. MMD is OK with batch_size=2. MMD can be trained with SGD, if using the unbiased estimator of MMD from the original "A Kernel Two-Sample Test" paper. So MMD can converge to the right distribution if using small minibatches. On the other hand, PMD does not have a known unbiased estimator of the gradient.

Reviewer 2



This paper presents the population matching discrepancy (PMD) as a better alternative to MMD for distribution matching applications. It is shown that PMD is a sampled version of Wasserstein metric or earth mover’s distance, and it has a few advantages over MMD, most notably stronger gradients and the applicability of smaller mini-batch sizes, and fewer hyperparameters. For training generative models at least, the MMD metric does suffer from weak gradients and the requirement of large mini-batches, the proposals in this paper therefore provides a nice solution to both of these problems. The small mini-batch claim is verified quite nicely in the empirical results. The verification of the stronger gradients claim is less satisfactory, since the MMD metric depends on the scale parameter sigma, it is essential to consider either the best sigma or a range of sigmas when making such a claim. In terms of having fewer hyper-parameters, I feel this claim is less well-supported, because PMD depends on a distance metric, and this distance metric might contain extra hyperparameters as well as in the MMD case. Moreover, it is hard to get a reliable distance metric in a high dimensional space, therefore PMD may suffer from the same issue of relying on a distance metric as MMD. On the other hand, there are some standard heuristics for MMDs about how to choose the bandwidth parameter, it would be good to compare against such heuristics and treat MMD as a hyperparameter-free metric as well. Overall I think the proposed method has the nice property of permitting small minibatch sizes therefore fast training. It seems like a valid improvement over large batch MMD methods. But the it still has the problem of relying on a distance metric, which may limit its success on modeling higher dimensional data.

Reviewer 3



The authors present PMD a population based divergence between probability distributions and show it is a consistent estimator of the Wasserstein distance. The estimator presented is conceptually simple and differentiable, which is a clear alllows training NN based models. The authors thoroughly compare PMD to MMD, which is the most prominent population based divergence in machine learning. The authors comment on the drawbacks of their method: exact calculation has cubic complexity, but propose the use of an approximation which has quadratic complexity, and show in their empirical results that this does not degrade statistical performance too much. The paper is well structured and written and includes references to previous work where due. The theoretical results seem correct. The experimental analysis is adequate. They compare PMD to MMD and other methods for domain adaptation and compare to MMD for generative modelling. I would have liked to see the method being used for generative modelling in domains with many modes. I wonder if PMD works when N is smaller than the number of modes. All things considered I think this is a good paper, that presents a possibly very useful method for comparing distributions.